# Evaluation of Spatiotemporal Characteristics of Lane-Changing at the Freeway Weaving Area from Trajectory Data

Pengying Ouyang [1,*] and Bo Yang [2]

1   Department of Transportation Engineering, School of Transportation, Southeast University, Southeast University Road #2, Nanjing 211189, China
2   School of Internet, Anhui University, Feixi Road #3, Hefei 230039, China
*   Correspondence: ouyangpengying@163.com

**Abstract:** Intensive lane-changing (LC) events are one of the great causes that make freeway weaving areas become bottlenecks. This study proposes an approach using vehicle trajectory data to investigate the spatiotemporal distributions of the number of LC events, void occupancies, and throughput variations at the freeway weaving area. Firstly, all LC events are extracted from the cleaned dataset and classified into four types according to the LC vehicles' origin–destination lanes and LC directions. Secondly, the time and space void occupancies are calculated using the kinematic theory. Thirdly, the throughput variations are identified with the oblique N-curve method. Finally, the spatial and temporal distributions of the LC events, void occupancies, and throughput variations are plotted to analyze their characteristics and relationships. The spatial distributions of different types of LC events indicate that most LC events occur at the surrounding area of the on-ramp entrance. Spatial distributions of time void occupancies show that the time void in the original lanes is quite small while that in the target lanes is much larger. Furthermore, the time void occupancies amplify downstream when considering vehicles traveling on the road. By comparing the temporal distributions of LC events, void occupancies, and throughput variations, there is a lag effect between the large value occurrences of space void occupancy and throughput reduction and that of the LC events, which can conclude a causal relationship between LC events and the occurrences of the space void occupancies and throughput reductions.

**Keywords:** lane-changing; void occupancy; throughput variation; spatiotemporal distribution

## 1. Introduction

The weaving area is an essential component of freeways, whose traffic flow conditions significantly influence the traffic efficiency and safety of the entire freeway system [1–3]. However, serving as the junction between freeways and ordinary roads, a large number of merging and diverging vehicles can only access or exit freeways through the weaving area. The merging and diverging vehicles need to cross over a distance within the weaving area to reach their target lanes. Such a process is referred to as the weaving operation [4]. Weaving operations involve extensive lane changes, which, coupled with limited road geometry boundaries of the weaving area, contribute to many discretionary lane changes of through vehicles. Such a high frequency of lane changes greatly impacts the traffic flow characteristics within the weaving area [5,6]. Therefore, it is essential to analyze the impacts of lane-changing (LC) behavior at the weaving segment on traffic flow in order to enhance traffic efficiency and safety at the weaving segment from the perspective of stable operation of traffic flow.

Several studies have been conducted to investigate LC mechanisms, such as intention detection and prediction [7,8], transition of the LC decision-making process [9,10], and LC risk evaluation and prediction [11–13]. Specifically, the study of LC intention detection and prediction involves monitoring the operational states of a vehicle, determining its LC intent,

and predicting its following driving states. The findings of such studies are advantageous for proactively avoiding LC risk and enabling coordinated cooperation between LC vehicles (LCVs) and their surrounding vehicles (SVs), thus reducing traffic congestion [14,15]. The study of the LC decision-making process arises due to the evidence that LC requires a period to complete, although LC is usually assumed as an instantaneous event in some macroscopic studies and simulations. Some researchers have divided the LC process into several stages, with one commonly used method being anticipating in the original lane and relaxation in the target lane [5,16,17]. Additionally, due to the constant influences from SVs on the operation of LCVs, the entire LC process may be prolonged and shortened according to the speed, spacing, and other factors that may impact LC [10,18]. Considering that LC is complex involving interactions with multiple vehicles, numerous studies have also investigated the conflict risk associated with LC [12,19]. Some researchers predict the likelihood of conflict risk during the LC process using pre-LC data [20,21], while others assess the consequences of LC accidents and analyze the causes using the accident-relevant traffic flow data [22,23].

In addition to the aforementioned research on LC behavior itself, numerous studies have conducted analyses of traffic flow characteristics influenced by lane changes [12,24,25]. It is generally acknowledged that LC maneuvers can trigger traffic oscillations and stop-and-go waves, and the oscillation waves could amplify and propagate upstream via vehicle platoons [26,27]. For example, ref. [28] developed a strategy-based LC model using observational data, and the results showed significant improvements in modeling LC maneuvers on freeways. Furthermore, many researchers have analyzed how LC maneuvers contribute to the oscillations and their propagation regularities [29,30]. For example, ref. [3] investigated how LC maneuvers interacted spatially and contributed to capacity drop at the merge, diverge, and weaving bottlenecks. Ref. [31] proposed an optimization strategy to control lateral flows upstream of the lane-drop location and to minimize the total travel time by maximizing the exit flows at the bottleneck.

The aggregation of lane changes can result in a reduction in road throughput capacity. This is due to the temporal and spatial voids in traffic flows created by lane changes, thereby diminishing the operational efficiency of vehicles. Therefore, there are many studies that have investigated methods for calculating the voids generated by lane changes and their relationships with throughput variations. For example, ref. [3] applied the kinematic formula along with graphic analysis to calculate void occupancy. The numeric simulation was used to explore the capacity drop under various traffic conditions and geometric characteristics. In their subsequent study, ref. [32] employed three car-following models along with graphic analysis and numeric simulation to investigate the combined effects of different acceleration, free-flow speed, and car-following behavior on traffic dynamics and throughput under a mixed traffic environment with automatic and regular vehicles.

In the field of connected and automated vehicles (CAVs), many studies have focused on collaborative LC trajectory planning in multi-vehicle scenarios [20,33,34]. For example, ref. [35] proposed a pre-cleaning strategy to improve the efficiency of emergency vehicles (EVs) by advising other vehicles to change lanes to clear a lane for the EVs. Ref. [36] analyzed the cooperative driving of connected and autonomous vehicles (CAVs) in non-signalized intersections and ramping regions by considering the time and location of performing LC maneuvers. Ref. [37] proposed a lateral LC control strategy for dynamic trajectory planning based on tracking the speed and acceleration of surrounding vehicles.

Recently, with the development of data collection devices and automatic extraction algorithms, an increasing number of scholars have employed high-precision trajectory data for the study of LC behaviors [38–40]. For example, ref. [41] presented a trajectory panning model for dynamic lane changes, utilizing time-independent polynomial curves to overcome unrealistic assumptions in existing models. This enables CAVs to adjust speeds and positions based on actual conditions, ensuring the efficiency and safety of lane changes. The effectiveness of this model was validated through real trajectory data. Ref. [42] extracted trajectory data from videos collected at 12 weaving segments to analyze

the effects of traffic flow characteristics and configuration elements of weaving segments on LC rates.

Based on the aforementioned literature review, it can be found that micro-level investigations into LC behavior primarily focus on individual behavior modeling, risk prediction for single LC maneuvers, and trajectory planning for single or multiple vehicle combinations [27,43]. From a meso- and macro-level perspective, numerous studies have analyzed the impact of lane changes on traffic flow, including the generation of temporal and spatial voids and throughput reductions [6,44]. However, there is a paucity of research that examines the influence of lane changes at different stages on the creation of voids in traffic flow on both the original and target lanes, along with their distribution characteristics on roads with special geometry. To this end, this study analyzes the spatial and temporal distributions of LC events and the corresponding disturbance in traffic flow characteristics at the freeway weaving area. The disturbance is represented by the time and space void occupancies and the throughput reductions. The LC events are categorized into mandatory LC events and discretionary LC events. Moreover, we consider the void occupancies in both time and space and calculate them on the original and target lanes for the anticipation and relaxation processes, respectively. We plot the spatial and temporal distributions of the LC events and the resulting void occupancies and throughput variations to analyze their disturbance and propagation characteristics in the weaving area.

The remainder of this paper is organized as follows. Section 2 introduces the data resource. Section 3 presents the methodology. Section 4 contains the results and discussion. The final section summarizes the study and proposes further work.

## 2. Studied Area and Raw Data

In this study, 45 min of video-based vehicle trajectory data were collected from the US Freeway 101 (US 101) by the Next Generation SIMulation (NGSIM) program. The data were recorded on 15 June 2005, from 7:50 a.m. to 8:35 a.m., and were segmented into three 15 min clips. Trajectories of all road users were extracted from the video data. The data information indicated that the traffic condition transferred from free-flow to congestion during the 45 min. Such a dataset could explain the general traffic characteristics, as the primary characteristic of weaving sections lies in the abundance of LC maneuvers. During the morning peak hour, not only are there more vehicles on the road, but lane changes also occur more frequently. This allows a clearer observation of the impact of LC behavior on traffic flow, encompassing both the spatiotemporal voids generated within the traffic flow and throughput variations of weaving sections. A summary of the trajectory components is presented in Table 1.

**Table 1.** The parameters and illustration.

| Parameter | Explanation |
|---|---|
| Vehicle_ID | Vehicle identification number. Unit: number. |
| Frame_ID | Frame identification number (ascending by start time). Unit: 1/10 s. |
| Total_Frames | The total number of frames in which the vehicle appears in this dataset. Unit: 1/10 s. |
| Global_Time | Elapsed time since 1 January 1970. Unit: milliseconds. |
| Local_X | The distance from the front center of the vehicle to the left-most edge of the section. Unit: feet. |
| Local_Y | The distance from the front center of the vehicle to the entry edge of the section in the direction of travel. Unit: feet. |
| Vehicle_Class | 1—motorcycle, 2—auto, 3—truck |
| Velocity | The instantaneous velocity of vehicles at this point. Unit: feet/second. |
| Acceleration | Instantaneous acceleration of vehicle at this point. Unit: feet/second. |
| Lane | Mainline: 1 to 5. Auxiliary lane: 6. On-ramp: 7. Off-ramp: 8. |
| Preceding_VID | Vehicle_ID of the lead vehicle of the subject vehicle. '0' represents no preceding vehicle, usually occurring at the end of the study section and off-ramp or in unsaturated traffic conditions. |
| Following_VID | Vehicle_ID of the lag vehicle of the subject vehicle. '0' represents no following vehicle, usually occurring at the beginning of the study section and on-ramp or in unsaturated traffic conditions. |

The study area is shown in Figure 1. The weaving area is marked as red. Two blue areas denote the basic freeway segments. The on- and off-ramps are marked as green. A right arrow indicates the traffic flow direction. The lane numbers are also marked in the figure. The length of the selected section is 2350 feet according to the longitudinal position of all vehicles in the dataset. The beginning section is denoted as 0. Along the traffic flow direction, the ranges of Lanes 7, 6, and 8 are 368.17 to 655.58 feet, 615.42 to 1350.05 feet, and 1309.76 to 1606.20 feet, respectively. These three sections cover each other over a distance of about 50 feet, possibly due to the angle of projection and the length of vehicles.

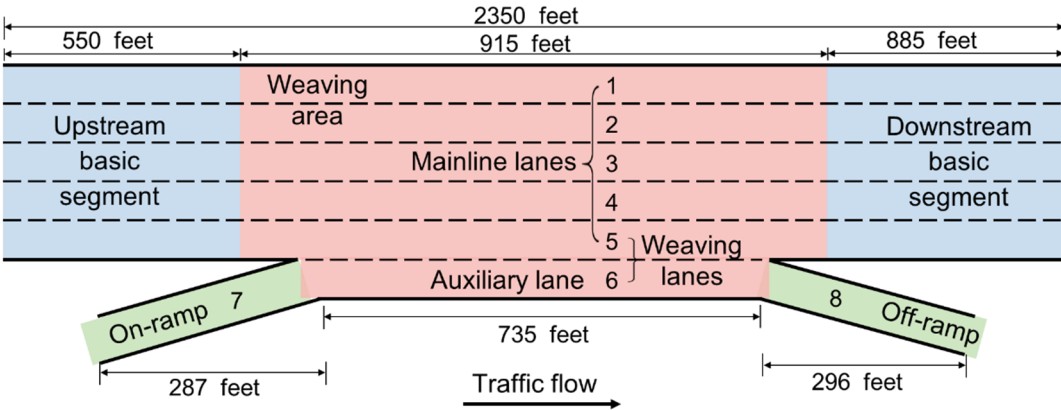

**Figure 1.** Diagram of the study area.

## 3. Methodology

The methodology applied in this study comprises the following four steps. Firstly, all LC events are extracted from the database and classified according to the vehicles' origin–destination and the directions of LC maneuvers. Secondly, the calculation of void occupancy in time and space is determined with the kinematic principle. Thirdly, throughput variations are examined by the oblique N-curve method. Finally, the spatial and temporal distribution of LC events, the void occupancy, and the throughput variation are analyzed, respectively, and the relationships among them are also discussed. The framework of the methodology is shown in Figure 2.

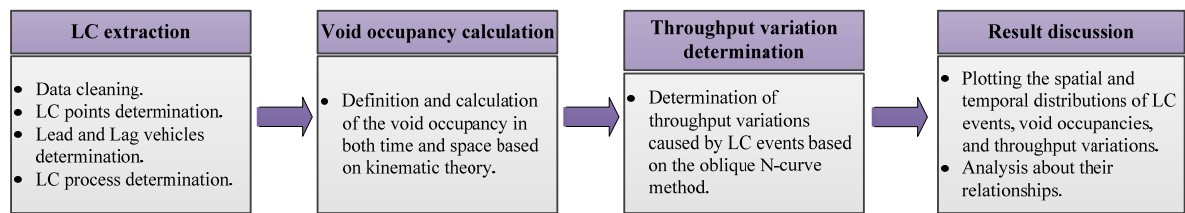

**Figure 2.** Methodological framework.

### 3.1. LC Extraction

LC extraction is proposed to extract the trajectory data of LC vehicles (LCVs) and their corresponding lead and lag vehicles in both the original lane and target lane during the LC process. Vehicle trajectory data are recorded as a series of spatial-temporal points of vehicles. The basic information of the trajectory data includes the vehicle ID (VID) and timestamp, as well as the vehicle's type, size, position, speed, and acceleration. Based on the basic information, several features can be further derived, such as the space and time headway, and the VIDs of the preceding and following vehicles of a subject vehicle.

Before extracting LC events, the raw data are cleaned by discarding the outliers, wrong messages, and duplications and then clustered with respect to vehicles. A particular vehicle in the dataset is defined by Vehicle_ID and Total_Frames, and the same Vehicle_ID can denote multiple vehicles. In this study, LC events are detected according to the four criteria

defined by [41], which are that the lateral distance of LCV is more than 2.2 m, the minimum lateral acceleration of the following vehicle is less than 0.07 m/s², the longitudinal distance of LCV and SV is less than 75 m, and the speed of all involved vehicles should be more than 1 m/s. An R program is developed to go through all points of a vehicle to recognize a change of lane number. The last point of an LCV in the original lane and the first point of the LCV in the target lane are recorded as BLCP and ALCP, respectively. A small number of vehicles driving on the lane markings confuses the automatic LC identification process and are discarded manually.

In this study, we used the definition of the LC process suggested by [45] to determine its beginning and ending points. As shown in Figure 3, points A and B, which are 1/4 lane width away from the lane marking, are respectively set as the beginning and ending points of the LC process. Thus, the anticipation process refers to the process of the LCV moving from A to C, while the relaxation process indicates the process of the LCV moving from C to B. The surrounding vehicles (SVs) of an LCV in the original and target lanes are illustrated in Figure 4. The lead and lag vehicles in the original lane before the LCV reaching the LCP are named as LeadB and LagB, respectively, while those in the target lane after the LCV passes through the LCP are named as LeadA and LagA, respectively. In addition, the trajectory data were collected at a ramp-weave weaving segment; see Figure 5. To better represent the LC characteristics, we classified the LC maneuvers into four types according to the origin–destination and LC direction of the LCV, namely, merge vehicle LC, diverge vehicle LC, through vehicle inside LC, and through vehicle outside LC, as shown in Figure 5.

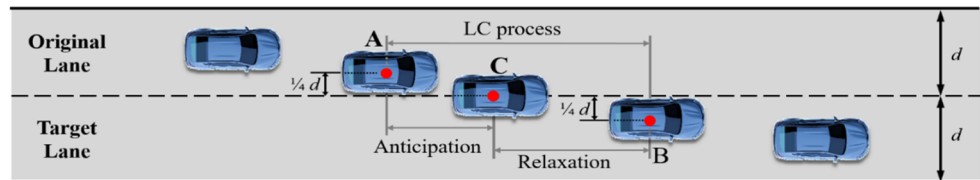

**Figure 3.** LC process illustration. Red dots are A, B, and C, marked in the center of vehicles.

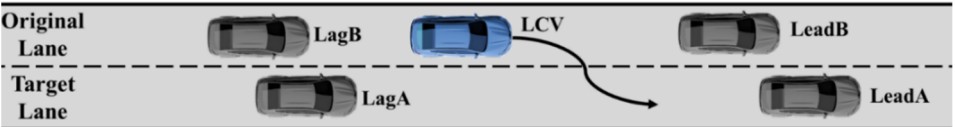

**Figure 4.** Position illustration of an LCV and its four surrounding vehicles.

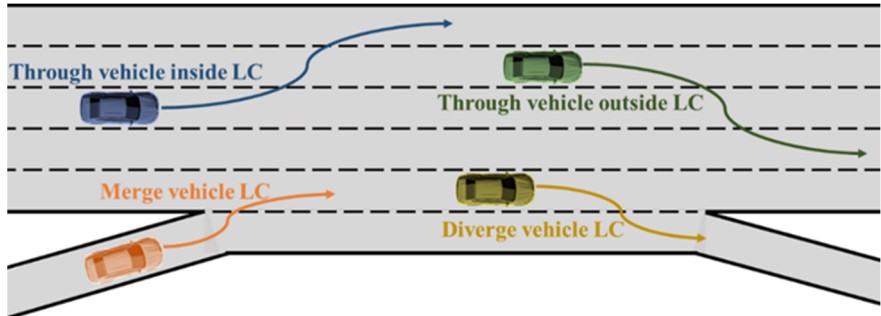

**Figure 5.** Illustration of four types of LC maneuvers.

### 3.2. Void Occupancy Calculation

LC events can trigger the void occupancy in front of LagB and LagA vehicles. When the traffic is not heavy and there are no vehicles following an LCV, the void occupancy is set at zero. In this study, we use the graphical method along with the kinematic theory mentioned in [3] to calculate the void occupancy induced by an LCV from the aspects of

space and time. Figure 6 illustrates the calculation of space and time void occupancies in the trajectory diagram with a space–time coordinate system. It should be noted that the lateral position of such a space–time coordinate diagram is fixed and only indicates one lane and the lines in the diagram show the vehicle's position change over time. Except for the LCVs, all vehicles are represented by black solid lines, with the assumption that the vehicles do not change lanes and travel at a constant speed. The solid lines are the actual trajectories generated by vehicles driving on the current lane. The LCV trajectories are represented by red/blue solid/dashed curves. The red solid curve is the actual trajectory generated by an LCV driving in the current lane while the red dashed curve is the actual trajectory generated by an LCV in the other lane. The blue dashed curve is the virtual trajectory of an LCV in the current and other lanes if it did not change lanes and still drove in the original lane with the same speed. In Figure 6, the vertical two-way arrow indicates the space void occupancy induced by an LCV at the time t = $t_1$, denoted as $O_{t1}$; the horizontal two-way arrow indicates the time void occupancy induced by the LCV at the location x = $x_1$, denoted as $O_{x1}$. The equations of the space and time void occupancies are as follows:

$$o_t = X_{LCVir,t} - X_{LCAct,t} \tag{1}$$

$$o_x = T_{LCAct,x} - T_{LCVir,x} \tag{2}$$

where $o_t$ is the space void occupancy at time t (unit: feet) and the $o_x$ is the time void occupancy at location $x$ (unit: second). $X_{LCVir/LCAct,t}$ is the location of the virtual/actual LCV trajectory at time t. $T_{LCAct/LCVir,x}$ is the time of the actual/virtual LCV trajectory at location x. The void exists only when the value of $o$ is larger than 0; otherwise, there is no void and its set value is 0. The blue dashed line was obtained by line regression based on the assumption that the LCV did not change lanes or speed. The red dashed lines in Figure 6a,b have the same longitudinal position and time as the actual trajectories in red solid lines in Figure 6a,b, respectively. The space and time void occupancies of multiple LC events are considered.

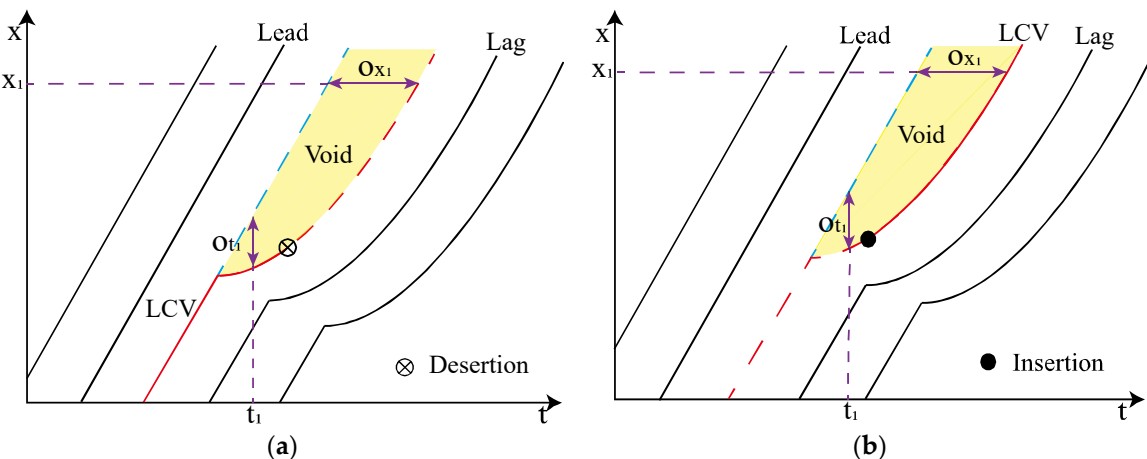

**Figure 6.** The principle of void calculation. (**a**) Void occupancy in the original lane, (**b**) Void occupancy in the target lane.

### 3.3. Throughput Variation Determination

A cumulative plot of the number of vehicles called the N-curve is commonly used to describe the traffic volume. However, due to its small variations in slope, it is difficult to reflect changes in throughput variations over a specific period. Therefore, a subtraction version called the oblique N-curve has been proposed to magnify the changes in the N-curve, allowing for a more intuitive observation of throughput variations based on the slope changes [46,47]. Figure 7 illustrates how the oblique N-curve can be obtained. Firstly, it is necessary to obtain the actual N-curve of the target section, see Figure 7a. Then, a

background flow, $q_0$, is determined and the assumed N-curve of the target section at a rate of $q_0$ is plotted, see Figure 7b. Finally, the oblique N-curve of the target section can be obtained by subtracting the data in Figure 7b from the data in Figure 7a. The background flow, $q_0$, needs to adhere to the following principle: slop variations are large enough that the variations can be distinguished by the naked eye [48].

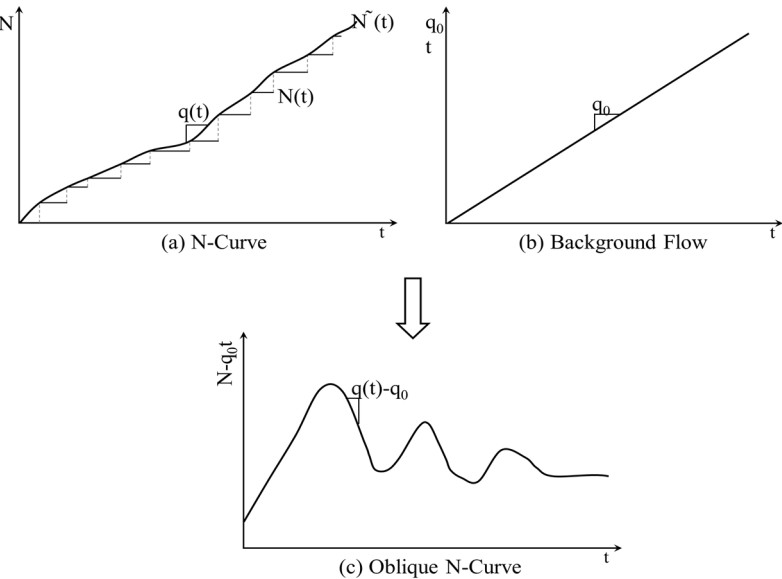

**Figure 7.** The illustration of oblique N-curve.

### 3.4. Spatial and Temporal Distribution Plotting

To better compare the similarities and differences of LC events, void occupancies in time and space, and throughput variations, their spatial and temporal distributions are plotted. The whole study area, including the weaving area and the basic freeway segments, is used for plotting the spatial distributions. The space unit for aggregating the number of LC events and the time void occupancies is determined based on the length needed for finishing an LC process, which is 50 feet in this study. Furthermore, as an LC event contains the anticipation process in the original lane and the relaxation process in the target lane, we take the last point of the LCV in the original lane to plot its location. The thermodynamic charts are used for exhibiting the spatial distributions. The time void occupancies in both the original and target lanes of each type of LC event are plotted to compare the differences in the anticipation and relations processes. As we are concerned about whether there are throughput reductions in the weaving area, the temporal distributions are plotted for all weaving sections. The time unit for aggregating traffic volume is set as 5 frames in this study.

## 4. Results and Discussion

### 4.1. Descriptive Statistics

A total of 6100 vehicles are recorded in the dataset, including 97% passenger cars, 1% motorcycles, and 2% trucks. Among all the vehicles, there are approximately 10% weaving vehicles, of which 6.4% are merging vehicles and 3.6% are diverging vehicles. In addition, three vehicles travel from the on-ramp to the off-ramp. The details of the number of these types of vehicles are summarized in Table 2.

**Table 2.** Details of all vehicles in the studied section.

| | Traveling OD | Through | Merge | Diverge | On- to off-Ramp |
|---|---|---|---|---|---|
| A total 6100 of tracked vehicles | No. of vehicles | 5475 | 399 | 223 | 3 |
| | Vehicle Type | Motorcycle | | Passenger car | Truck |
| | No. of vehicles | 45 | | 5918 | 137 |

After data cleaning, 1382 LCVs were detected in the dataset, including 399 merge vehicles, 223 diverge vehicles, and 760 through vehicles. As some LCVs change lanes more than once, a total of 1822 lane changes are identified. Among them, the number of through vehicle inside LC, through vehicle outside LC, merge vehicle LC, and diverge vehicle LC are 638, 267, 655, and 262, respectively, as shown in Table 3. In Table 3, "# of LC for an LCV" means the number of LC maneuvers conducted by the same LCV, and "# of LC events" refers to the frequency of the specific type of LC events. The sample sizes of four types of LC events are not well balanced, as the numbers of through inside and merging LC events are about three times as many as the through outside and diverging LC events. A close look at the data reveals that most LCVs change lanes once, accounting for 76.7% of the total LCVs, especially through outside LCVs and diverging LCVs. The remaining LCVs may change lanes two times or more to gain a better driving environment. Due to the calculation of void requiring speed, Table 4 presents the speed information. The data in Table 4 are obtained from the average speed of each vehicle passing through the weaving segment. From Table 4, the speed of through vehicles is unexpectedly the lowest, especially for through vehicles without changing lanes, with an average speed of approximately 30 feet/s. Both diverging and merging vehicles exhibit higher speeds, with average speeds around 40 feet/s.

**Table 3.** The number of different types of LC events.

| | # of LC for an LCV | Inside | 1 | 2 | 3 | 4 |
|---|---|---|---|---|---|---|
| Through LC: 905 Through LCV: 760 | # of LC events | 638 | 437 | 142 | 51 | 8 |
| | # of LC for an LCV | Outside | 1 | 2 | 3 | 4 |
| | # of LC events | 267 | 206 | 40 | 21 | 0 |
| Merging LC: 655 Merging LCV: 399 | # of LC for an LCV | 1 | 2 | 3 | 4 | 5 |
| | # of LC events | 205 | 198 | 129 | 88 | 35 |
| Diverging LC: 252 Diverging LCV: 223 | # of LC for an LCV | 1 | 2 | 3 | 4 | 5 |
| | # of LC events | 211 | 16 | 0 | 1 | 1 |

**Table 4.** Speed information of vehicles with different ODs (Unit: feet/second).

| | Type | Mean | Min | Max | Std |
|---|---|---|---|---|---|
| Through vehicles | Total | 32.21 | 16.07 | 60.96 | 8.84 |
| | With LC | 34.02 | 16.85 | 59.48 | 9.47 |
| | Without LC | 31.84 | 16.06 | 60.96 | 8.66 |
| Merging vehicles | | 39.22 | 18.64 | 64.27 | 10.22 |
| Diverging vehicles | | 42.63 | 17.54 | 56.98 | 7.65 |

*4.2. Spatial Analysis of LC Events*

The spatial distributions of four types of LC events are displayed in Figure 8. The color of each brick, ranging from green (R: 99, G: 190, B: 123) to red (R: 248, G: 105, B: 107), indicates the number of LC events occurring on the brick. The length of each brick is 50 feet (mentioned in Section 3.4) and the width is the same as the lane width. As the location of an LC event is defined by the last point of the LCV in the original lane in this study,

there must be one lane without any LC events when plotting the spatial distributions of LC events with one single LC direction. For example, in Figure 8b, LCVs change lanes from the outside lane to the inside lane, so Lane 1 cannot be the original lane for any LCV and it is blank in the figure. To better identify the LC event locations in the geometric diagram of the weaving area, two black dashed lines are used to separate the freeway basic segments from the weaving area, and two blue dashed lines are used to divide the weaving area into the upstream, middle, and downstream segments. The weaving area is separated from the basic freeway segments by black dashed lines and divided into the upper, middle, and lower segments by blue dashed lines. The area with a high frequency of LC events is highlighted by a purple box.

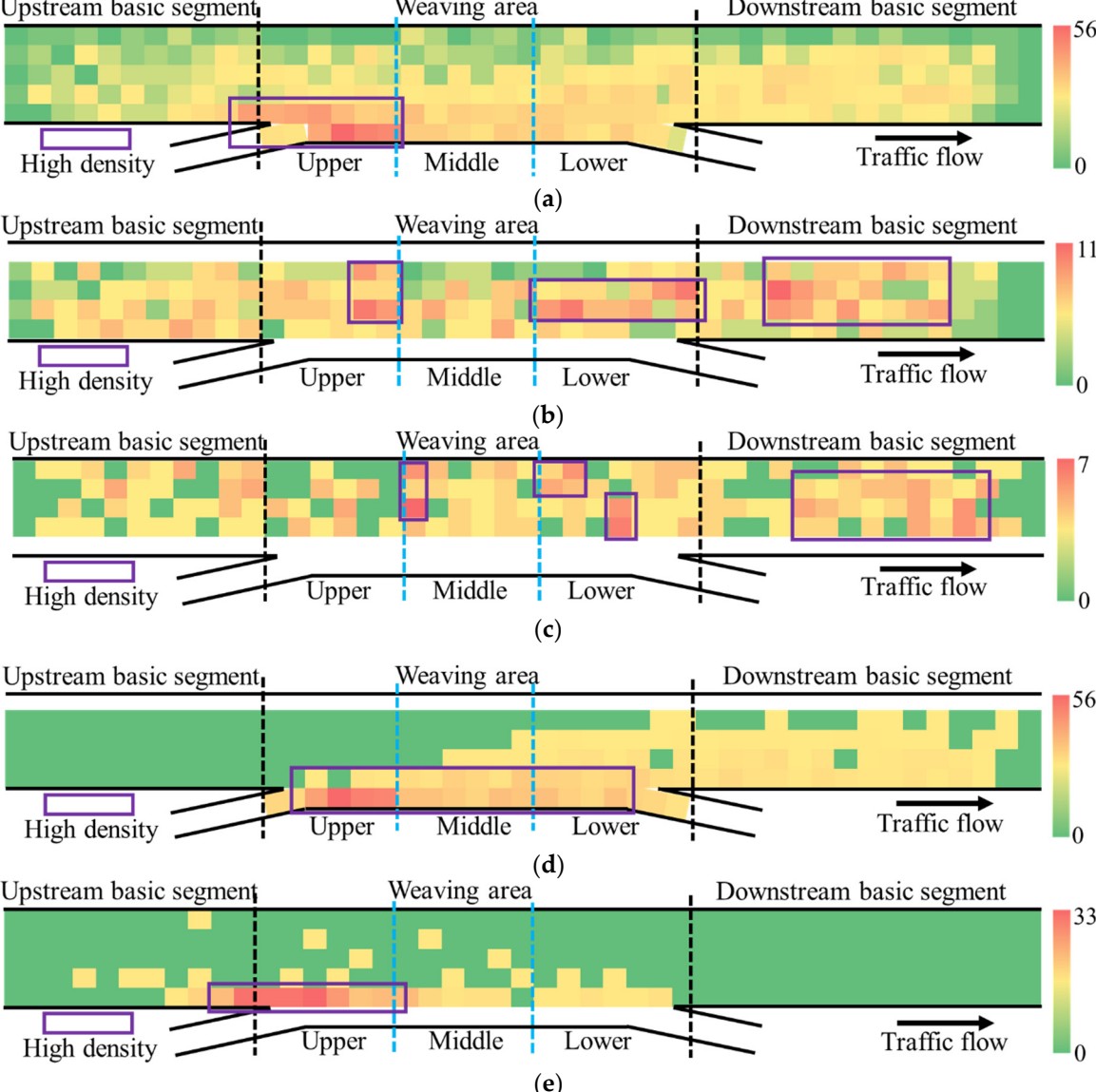

**Figure 8.** The spatial distributions for different types of LC events. (**a**) Number of total LC events, (**b**) Number of through vehicle inside LC events, (**c**) Number of through vehicle outside LC events, (**d**) Number of merging vehicle LC events, (**e**) Number of diverging LC events.

From the distribution of total LC events shown in Figure 8a, most LC events occurred on the auxiliary lane (No. 6) and the outermost lane (No. 5) in the weaving area. The highest LC frequency can be seen around the entrance of the on-ramp, mainly by merging vehicles moving to the mainline lane immediately after entering the on-ramp and diverging

vehicles preparing to join a different traffic stream by moving into the auxiliary lane before reaching the entrance. This observation is consistent with [49]. Fewer LC events were observed on the middle and outside lanes.

The spatial distributions of through vehicle inside LC events and through vehicle outside LC events are displayed in Figure 8b,c. The high frequency of both types of LC events are located at a similar area, i.e., around the left blue dashed lines and in the downstream basic segment. A high frequency of LC events conducted by through vehicles in the lower part of the weaving areas may be triggered by the large number of mandatory LC maneuvers conducted by weaving vehicles, as through vehicles need to make room for weaving ones. As for the high-frequency LC events in the downstream freeway basic segment, this may be caused by the through vehicles' self-organization after passing the bottleneck [50,51]. However, Figure 8b,c indicate that the area with a low frequency of through inside LC events is the middle part of the weaving area while the area with a low frequency of through outside LC events is the upper part of the weaving area. This is created by the through vehicles in the upper section of the weaving area moving away from the inside lane into the middle and outside lanes to make room for merging vehicles.

The areas highlighted by the purple rectangles in Figure 8d,e match the ones with high LC frequency in Figure 8a. In Figure 8d, most merging vehicles enter the mainline, i.e., changing lanes from No. 6 to No. 5, in the upper section of the weaving area. Additionally, many of them continue to change lanes to the middle lanes, resulting in the yellow fan-shaped area from the middle to the lower section of the weaving area in Figure 8d. This phenomenon is consistent with that of [52]. The number of diverging vehicles in the dataset is small. As can be seen in Figure 8e, most of them were already on the inside lane when reaching the weaving area, so they moved to the auxiliary lane as soon as they entered the weaving area. In both cases, there were no LC events on the inside lane. Furthermore, from the maximum value of the color bars in Figure 8b–e, the LC events made by weaving vehicles are concentrated in one area, the entrance of the on-ramp. Those made by non-weaving vehicles are dispersed within the whole study area.

*4.3. Spatial Analysis of Time Void Occupancy*

The time void occupancy induced by each LC event was calculated at multiple sub-sections. Starting from an integer value near Local_Y of the LCP, the segment in original and target lanes involved in an LC event were divided into 50-foot sections, the same length as each of the bricks in Figures 9 and 10. The entire area was divided into upstream freeway basic segment, weaving area, and downstream freeway basic segment by two black dashed lines. The weaving area was further separated into upper, middle, and lower segments by two blue dashed lines. The brick colors range from green (R: 99, G: 190, B: 123) to red (R: 248, G: 105, B: 107), representing the increase in the time void occupancy (unit: second). The time void occupancy in each brick is the sum of the void generated by all LC events occurring on the brick. It should be noted that the location of time void occupancy in the original lane is coordinated with the LCV's following vehicle, while that in the target lane is coordinated with the LCV. Furthermore, there must be one empty lane in Figures 9 and 10b–e, as the innermost/outmost lane could not be the original lane of the LC events to the inside/outside lanes and the innermost/outmost lane could not be the target lane of the LC events to the outside/inside lanes. The reason for choosing the position of coordinated vehicles is because they are the first ones in traffic flow to be affected by the time void occupancy. As the void is generated during the whole LC process when the vehicle goes downstream, it seems that the time void transmits downstream in the spatial distribution graphs because only the time void generated directly by the LCVs was considered in the spatial distribution analysis. Figure 6 shows the effect of time void on traffic flow propagated upstream over time, which is consistent with [53,54].

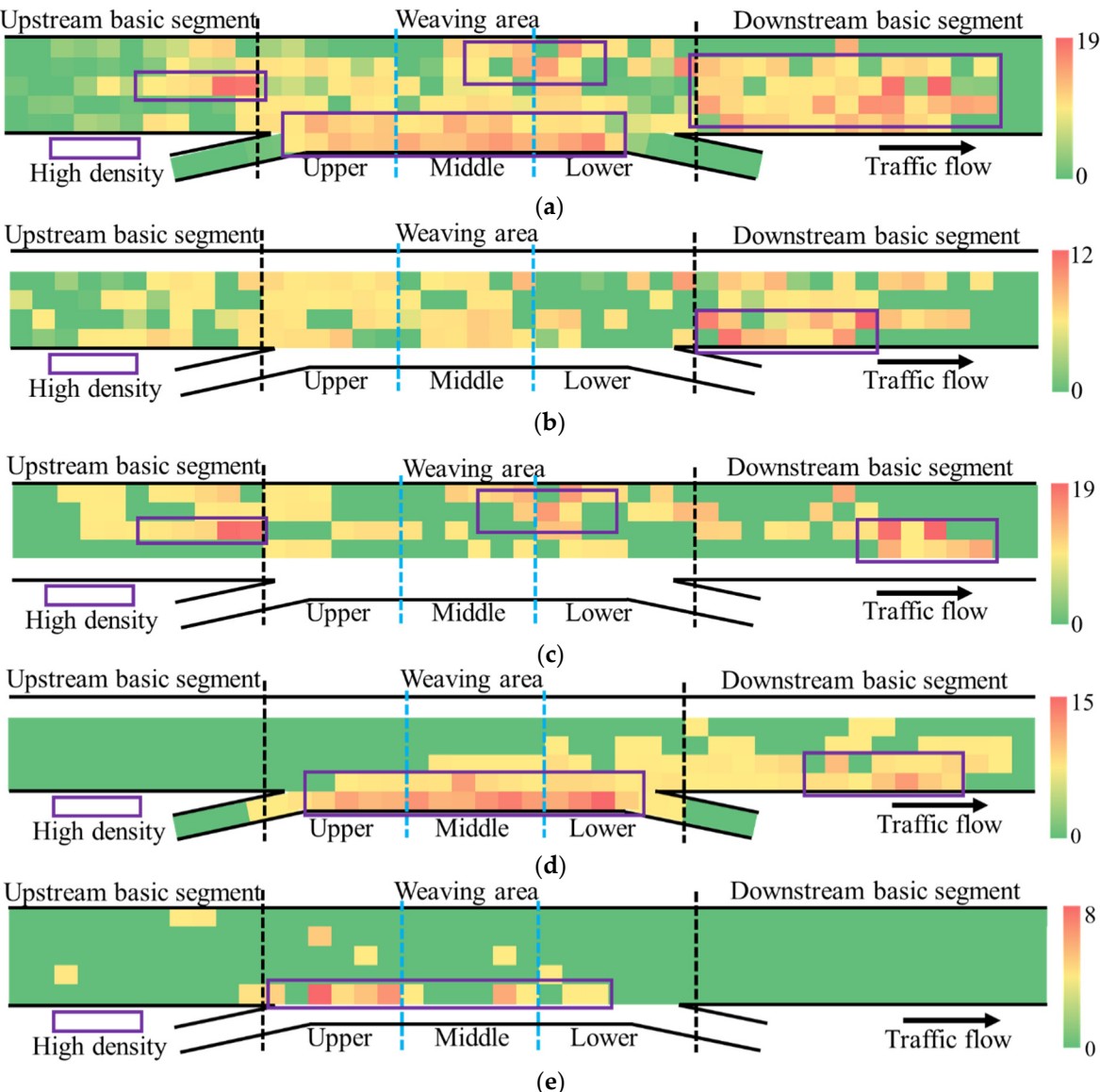

**Figure 9.** The spatial distributions of time void occupancy in the original lanes. (**a**) Time void occupancy generated by total LC events, (**b**) Time void occupancy generated by through vehicle inside LC events, (**c**) Time void occupancy generated by through vehicle outside LC events, (**d**) Time void occupancy generated by merging vehicle LC events, (**e**) Time void occupancy generated by diverging vehicle LC events.

The spatial distributions of time void occupancy generated by different types of LC events in the original lanes are displayed in Figure 9a–e using thermodynamic diagrams, while those in the target lanes are displayed in Figure 10a–e. It can be seen that the time void occupancy generated in the target lanes is much bigger than that in the original lanes. The maximum values of one brick unit of different types of LC events are similar among Figure 9b–e but vary significantly among Figure 10b–e. The spatial distributions of time void occupancy in the original lanes appear to be random and irregular. Moreover, the whole time void occupancy in the original lanes accounts for 8.21% of the total; hence, it can be ignored when considering the macroscopic level. The spatial distributions of time void occupancy in the target lanes are consecutive and the value decreases gradually from the maximum point (a red brick) to its periphery. All these characteristics of spatial distributions, both in original and target lanes, are consistent with the traffic situation stated in previous works. For example, [55,56] point out that the gap left by an LCV in the

original lane can be filled immediately by its following vehicle or another LCV moving in from the adjacent lane, which reduces the time void, while on the target lane, the following vehicle of an LCV often creates an additional gap when their speed or acceleration or both are lower than that of the LCV.

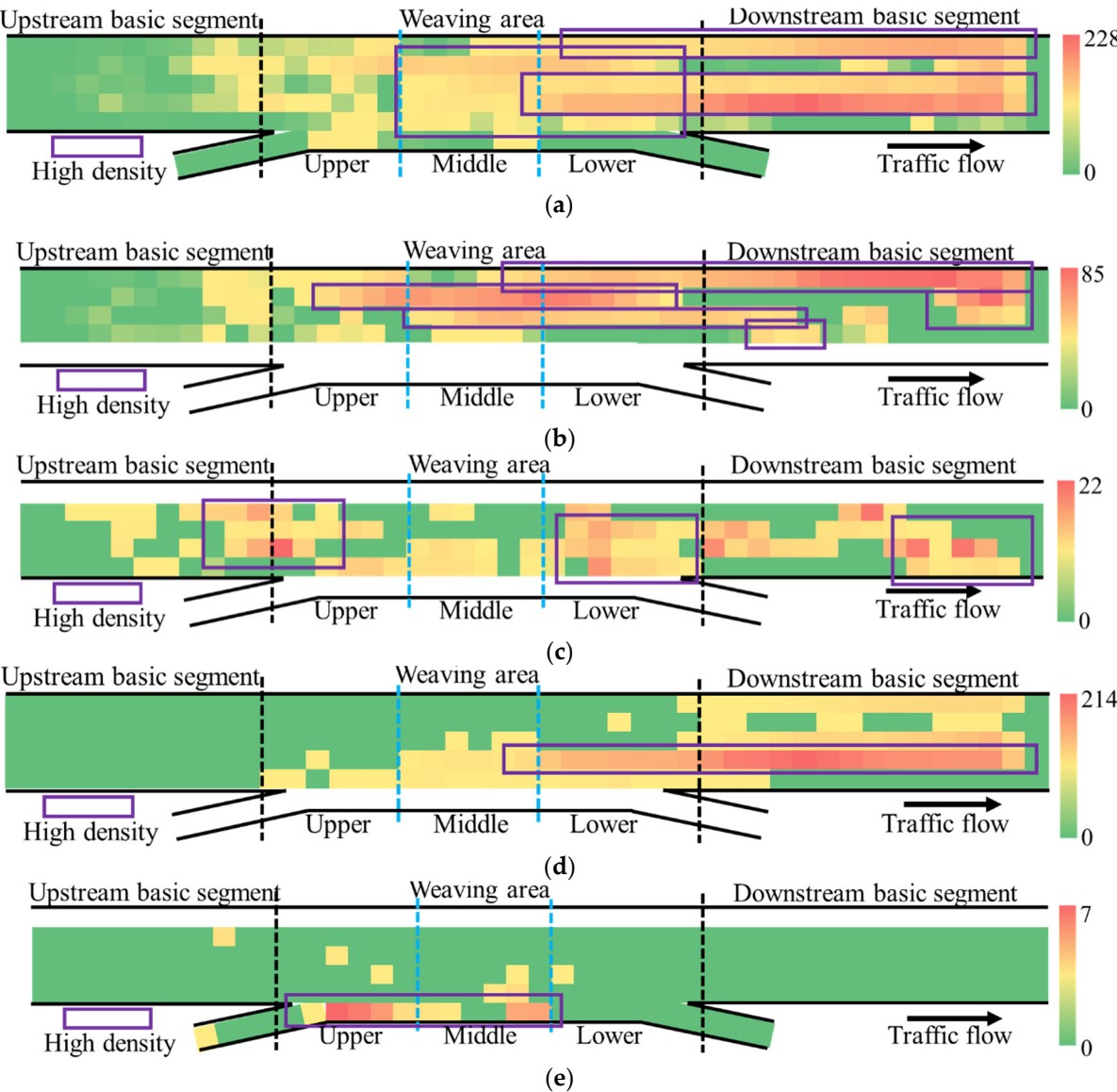

**Figure 10.** The spatial distributions of time void occupancy in the target lanes. (**a**) Time void occupancy generated by total LC events, (**b**) Time void occupancy generated by through vehicle inside LC events, (**c**) Time void occupancy generated by through vehicle outside LC events, (**d**) Time void occupancy generated by merging vehicle LC events, (**e**) Time void occupancy generated by diverging vehicle LC events.

In Figure 9d,e, time void occupancy in the weaving lanes (Lanes 5 and 6) is mainly produced by weaving vehicles. The diverging vehicles produce little time void occupancy (Figure 9e). Besides the weaving lanes, the merging vehicles tend to create large time void occupancy in the downstream freeway basic segment (Figure 9d). Figure 9b,c show that through vehicles tend to improve their driving environment and make several LC maneuvers after passing through the weaving bottleneck; hence, a large time void occupancy is shown in the downstream basic segment in Figure 9b,c, which corresponds to the LC frequency shown in Figure 8b,c.

The time void occupancies induced by the through vehicle inside LC events (Figure 10b) and merging vehicle LC events (Figure 10d) contribute to most of the total value shown in Figure 10a. Especially, the two orange-to-red strips at the downstream freeway basic segment in Figure 10a are consistent with the red strips shown in Figure 10b,d, respectively. In contrast, the other two types of LC events generate smaller time void occupancies. In Figure 10a, most total time void occupancies are distributed in the downstream freeway basic segment and the lower and middle parts of the weaving area, as the through traffic was interrupted by the merging vehicles, leading to large void occupancies. From the overall view, a large void occupancy in the downstream and a small one in the upstream were also observed. In the downstream, the majority of merging vehicles have entered the mainline traffic and some diverging vehicles have not arrived at the off-ramp, so the traffic in the area can be relatively chaotic, and a large number of time void occupancies appear.

As the location of time void occupancy is identified via the trajectories of the LCVs in the target lane and their following vehicles in the original lane, the spatial distributions of time void occupancy are shown in Figures 9 and 10 and those of LC event frequencies shown in Figure 8 are related. From the whole view, the spatial distributions and values of the merging void and diverging void in the original lane along with weaving vehicles have the highest similarity among these three group graphs. The spatial distributions of the merging LC events and their corresponding time void occupancies in both the original and target lanes are fan-shaped, like a sector radiating outwards from the gore of the on-ramp in the weaving area (see the fourth graph in Figures 8–10). The spatial distribution of the diverging LC events and their time void occupancy are similar in two ways: they are small in value and scattered sporadically near the entrance of the on-ramp. Furthermore, there is a commonality between the spatial distribution of the merging and diverging LC events and their time void occupancies, which is that the location of the highest value of the time void occupancies in the downstream mirrors the location of the highest number of LC events. The through vehicle inside and through vehicle outside LC events are equally distributed in the whole area, and so are the time void occupancies generated by them. The bricks with a high value, which are in red in the (b) and (c) graphs in Figures 8 and 9, are located downstream of the red bricks in Figure 8b,c. There are no obvious features in their spatial distributions. This may be the reason that the LC events conducted by through vehicles are discretionary.

### 4.4. Temporal Analysis

This section shows the temporal distributions of the throughput variations at various sections, the LC events, the total space void occupancy, and their relationships. According to Section 3.4, the sections with the 50-foot interval from 550 feet to 1350 feet are chosen to inspect the throughput variations and are denoted as S550 to S1350 in Figure 11. To ensure the accuracy of the oblique N-curve, only 35 min data are used in the time distribution analysis after discarding the first and last 5 min intervals of the whole period.

The number of vehicles passing through every chosen section was recorded. For each section, there were three cases to be considered, the full section, the mainline section with five lanes, and the ramp section consisting of the auxiliary lane and the ramps. The background traffic volume $q_0$ is determined by the average values, which are 11, 10, and 1 vehicle/5 s for the full section, the mainline section, and the ramp section, respectively. Figure 11 shows the oblique N-curves for all selected sections under three cases. The abscissa is a total of 420 intervals with a unit of five frames. The ordinate represents the differences between the cumulative real traffic volume and the cumulative background traffic volume. The slope of each curve is the value of the actual capacity over a period minus $q_0$. Since the slope changes of all oblique N-curves in Figure 11 are significant enough to be directly observed, the illustrated oblique N-curve is reasonable and feasible. Thus, the chosen $q_0$s are appropriate.

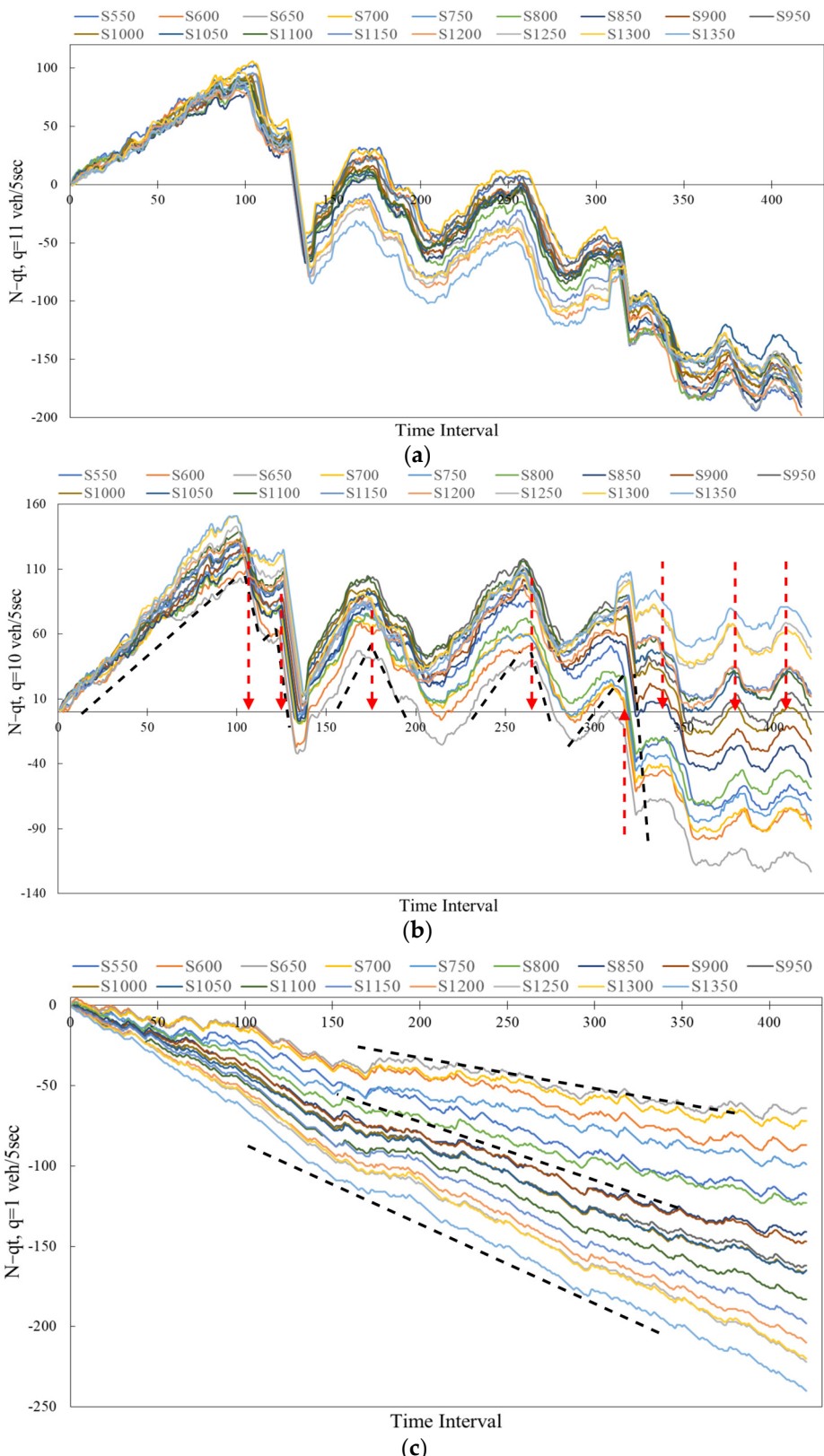

**Figure 11.** The oblique N-curves for various sections. (**a**) The oblique N-curves of full sections, (**b**) The oblique N-curves of the mainline sections, (**c**) The oblique N-curves of the ramp sections. Dashed lines are the approximate trend lines of the slope of curves.

Figure 11a,b reveal that the throughput variations of the full sections and the mainline sections are similar. Furthermore, the significant fluctuations at different locations are also in the same trend, i.e., centered near the points indicated by the red arrows. The difference between the two subgraphs is that the curves at different locations become dispersed in the mainline section case, but remain converged in the full section case. In Figure 11c, the slopes of all curves are almost constant, indicating that the throughputs of the ramp sections at different longitudinal locations remain stable. Different slopes of curves at different sections mean that each section has a different steady throughput rate, which is why the curves in Figure 11b become dispersed over time.

In Figure 11a, the throughput rates of the full sections are about 12 veh/5 s during the first 100 intervals and then decrease sharply to 9 veh/5 s. The throughput rates rise imperceptibly in the following 20 intervals, then drop to 0 and remain there in the next 10 intervals, which indicates the road is jammed and no vehicle is moving. This explains why the curves of different sections are only superposed in the green circles in Figure 11a. Then, these curves go through three large fluctuations from the 140th to 320th 5 s interval. The reductions of the three changes were about 8, 5, and 13 veh/5 s, respectively. In the last descent process, the sections experienced congestion again, as the throughput rates dropped to zero. Finally, the curves experienced three short-lived and small fluctuations, so they could be seen as natural fluctuations of traffic flow. The trends of throughput variations of the mainline sections are similar to the above description for the full sections. The only difference is that the discharge rates for all the ramp sections at different locations are stable; the traffic flow observed at the different mainline sections is the result of those of the full sections minus the discharge rate of the ramp sections.

Figure 12 is the time distribution of all LC events that occurred at the same time intervals. The bars with high LC frequencies are marked in red, which corresponds to the large fluctuations of oblique N-curves shown in Figure 11a. Comparing the time intervals of the red bars in Figure 12 with the time intervals pointed by the red arrows in Figure 11, it can be concluded that the time points of the throughput reductions lag behind those of the high LC frequencies. This hysteresis effect is consistent with previous research on how LC influences oscillation formation and discharge rate [57,58]. A common feature of these red-bar clusters is that they have a peak bar with the highest value. As it takes time to form a traffic jam, the congestion highlighted by two green circles in Figure 11a may not be the result of the one red-bar cluster but that of the continuous high LC frequencies during the periods before the congestion, which are framed by black dashed lines in Figure 12.

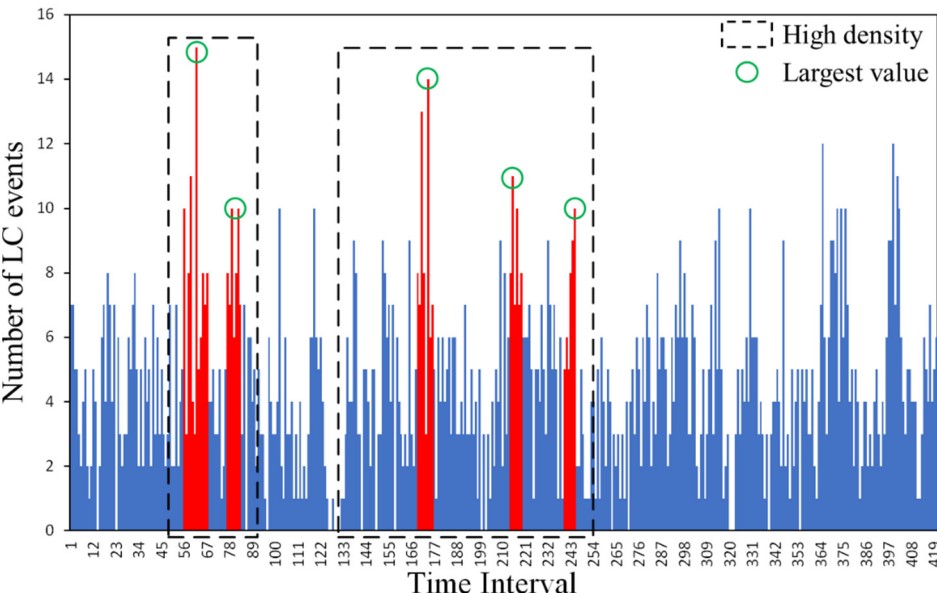

**Figure 12.** The time distribution of LC frequency during 5~40 min.

Figure 13 displays the time distribution of the space void occupancy using the stacked plot. The red, black, blue, and green shaded areas correspond to the space void occupancies generated by the through vehicle inside, through vehicle outside, merging vehicle, and diverging vehicle LC events, respectively. The sum of the space void occupancies is represented by the area under the black solid line. From Figure 13, the total space void occupancies are mainly composed of those generated by through vehicle inside and merging vehicle LC events. The space void occupancies produced by two other types of LC events are too small to be considered. An interesting finding from Figure 13 is that the space void occupancies created by discretionary LC maneuvers conducted by through vehicles are much larger than those created by mandatory LC maneuvers made by weaving vehicles. The five clusters with high space void occupancies framed by the purple boxes correspond to the five red-bar clusters shown in Figure 12 and the five large fluctuations shown in Figure 11a. Comparing the time distributions presented in Figures 11–13, both the occurrences of large space void occupancies and the throughput reductions lag behind the peak LC-event frequencies. This is because the void occupancy and the throughput reduction are produced by LC events over a period of time. There is no obvious connection between the time points of throughput reduction and the peak space void occupancies. There is hardly any space void occupancy after the last peak value, proving that the last three small fluctuations in Figure 11a are the usual fluctuations. Although there were many LC events in the last period, the traffic is not heavy compared with the previous duration and many suitable gaps could be utilized by LCVs. Therefore, no space void occupancy is generated.

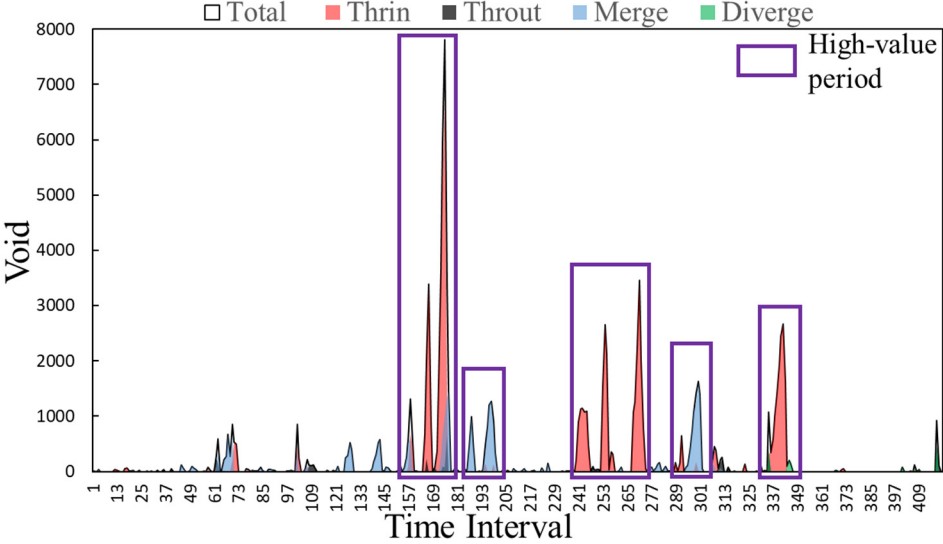

**Figure 13.** The time distribution of space void occupancy (unit: feet).

*4.5. Discussion and Suggestions for Practice*

This section encapsulates pivotal observations from the above analysis and proposes corresponding recommendations aimed at improving traffic safety and efficiency in the weaving segment. The delineation is as follows:

1.  The majority of merging and diverging vehicles tend to change lanes near the merge triangle area at the entrance of the weaving segment (Figure 8d,e). Specifically, lane changes made by diverging vehicles are more concentrated around the merge triangle area and most of them are distributed in the upper part of the weaving segment, with a few extending downstream on the outermost mainline lane. Conversely, lane changes of merging vehicles exhibit a distribution with the merge triangle as the highest point, gradually decreasing towards downstream and inner lanes, forming a fan-shaped distribution. Such a behavioral pattern for lane changes arises from the inclination of weaving vehicles to expeditiously effectuate lane changes toward

their target direction upon ingress into the weaving segment. This strategic maneuver facilitates weaving vehicles to complete weaving operations smoothly and allows them to assimilate into more optimal lanes within the confined space of the weaving segment.

In response to this behavioral tendency, the installation of signs upstream of the weaving segment and guided lane marking within the weaving segment would help to disperse lane changes more evenly across the entire length of the weaving segment, rather than concentrating at the entrance.

2. Most lane changes of through vehicles congregate in the midsection of the weaving segment and the downstream basic segment (Figure 8b,c). This predilection is an outcome of the influence exerted by weaving traffic on the lane change patterns of through traffic. Within the weaving segment, through vehicles change lanes to make space for weaving operations. After passing through the weaving segment, many through vehicles also make lane changes to search for a better driving condition.

To reduce discretionary lane changes of through vehicles, repetitive signboards upstream of the weaving segment can remind through vehicles to use the inner lanes. This can minimize the mutual influence between through traffic and weaving traffic, thereby enhancing traffic efficiency and safety.

3. It can be seen from the above analysis that lane changes only generate sporadic and dispersed temporal voids in traffic flow on the original lane (Figure 9). These temporal voids are inconsequential in magnitude when compared with those engendered on the target lane.

4. The time void in traffic flow on the target lane is primarily caused by through inside LC events and merging vehicle LC events (Figure 10a,d,e). This is because the quantity of these two types of lane change is approximately three times that of the other two types of LC events. While the number of merging vehicles is slightly more than diverging vehicles, the main difference in the number of lane changes is due to almost half of the merging vehicles making two or more lane changes, whereas around 95% of diverging vehicles make only one lane change. This indicates that diverging vehicles are well prepared for weaving operations when they are upstream of the weaving segment. Meanwhile, merging vehicles, after completing the weaving operation, are eager to change lanes towards the inner side within the weaving area to obtain a better driving environment.

To address this situation, reminders in navigation software can advise merging vehicles not to congest toward the inner side in traffic congestion but rather to promptly exit the weaving area. This can prevent exacerbation of the complex traffic flow conditions within the weaving segment.

5. Analysis from Section 4.4 reveals that a large number of lane changes in a short period can lead to traffic breakdown (Figures 12 and 13). Therefore, it is necessary to disperse lane changes both in terms of time and space to allow traffic flow to pass through the weaving segment more safely and smoothly. In addition to the previously mentioned methods of setting up signs and lane markings, implementing navigation system reminders can guide drivers to choose appropriate times for lane changes.

## 5. Conclusions

This study analyzes the spatial and temporal distributions of LC events and the corresponding disturbances in the freeway weaving area. The analysis contains four steps, namely, extracting and classifying the LC events, calculating the void in the anticipation and relaxation processes, identifying the throughput reductions, and analyzing the spatial and temporal characteristics of LC events and void occupancies and throughput variations. The US101 trajectory dataset in NGSIM is used for the case study. Data processing is conducted in R. The graphic method is applied to determine the void occupancies in time

and void. The oblique N-curve method is used to demonstrate the throughput variations. The following conclusions were made from the study:

(i)  The discretionary LC events from through vehicles take about 50% of the total LC events, which are much more numerous than the mandatory LC events in the weaving area. Both merging and diverging vehicles change lanes upon reaching the weaving area, i.e., the vicinities connecting the entrance of the on-ramp and the freeway mainline. After passing through the weaving section, both through and merging vehicles would like to change lanes again to improve their driving environment.

(ii)  The number of time void occupancies in the original lanes are less than 1/10 of those in the target lanes. Furthermore, the distribution of time void occupancies in the original lanes is scattered and irregular, while that in the target lanes is similar to the distributions of LC events.

(iii)  The diverging vehicle LC events produce fewer spatial and time void occupancies than the merging vehicle LC events. Through vehicle inside LC events generate more space void occupancies but fewer time void occupancies than merging vehicle LC events.

(iv)  From the temporal distributions, the occurrences of throughput reductions and high space void occupancies lag behind the occurrence of peak LC events, which illustrates that there is a causality between LC events and the occurrences of the space void occupancies and throughput reductions. From the spatial distributions, the locations of large time void occupancies are downstream of the locations of peak LC events.

This study and the above findings bridge the research gaps and contribute to the field of LC research in the following aspects: (i) analyzing the differences between the mandatory LC events and discretionary LC events conducted by weaving and non-weaving vehicles, respectively; (ii) modeling the time and space void occupancies and throughput variations in the anticipation and relaxation processes separately; (iii) exploring the propagation regulations of void occupancy under the weaving area context rather than only considering the x-t trajectory plot.

There are several limitations of this study. Firstly, due to the limited precision of the NGSIM dataset [59,60], such as unreasonable acceleration and the deviation of projection by using roadside cameras with angles, more details about LC events cannot be examined elaborately. Future studies could be conducted on modeling and predicting the space and time void occupancy generated by a single LC maneuver and multiple LC maneuvers from detailed data. Secondly, more weaving areas with different geometric designs should be analyzed to compare the influences of geometric characteristics on LC disturbances. Thirdly, more accurate models of the space and time void occupancies are needed to analyze the safety performance of LC maneuvers between LCVs and SVs. Further studies could be conducted to develop and predict the safety profiles between LCVs and SVs.

**Author Contributions:** Conceptualization, P.O.; Methodology, P.O.; Formal analysis, P.O.; Writing—original draft, P.O.; Writing—review & editing, B.Y.; Funding acquisition, B.Y. All authors have read and agreed to the published version of the manuscript.

**Funding:** This research is supported by the Natural Science Research Project of the University in Anhui Province of China (KJ2021A0019). The authors would like to thank the editor and the reviewers for their constructive comments and valuable suggestions to improve the quality of this article.

**Institutional Review Board Statement:** Not applicable.

**Informed Consent Statement:** Not applicable.

**Data Availability Statement:** This study used an open data, which can be found at https://ops.fhwa.dot.gov/trafficanalysistools/ngsim.htm (accessed on 14 October 2023).

**Conflicts of Interest:** The authors declare no conflict of interest.

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
