# Peer review of "Evaluation of Spatiotemporal Characteristics of Lane-Changing at the Freeway Weaving Area from Trajectory Data"

_sustainability, doi:10.3390/su16041639_

Round 1
Reviewer 1 Report
Comments and Suggestions for Authors
I am glad to review this manuscript entitled “Evaluation of Spatiotemporal Characteristics of Lane-changing at the Freeway Weaving Area from Trajectory Data”. In this manuscript, the authors analyzed the data well. However, I think cited references, extra information for methodology, and interpretation of data are not sufficient in the manuscript. Please refer to the following questions and checkpoints.
○ Major questions
#1. In the introduction, there are no references to arguments(L77-86)
#2. Since the explanation of the LC event criteria is closely related to this study, it would be better to disclose it in the main text (L141-143).
#3. There is no mention of specific research methods (what packages did they use in the R program?).
#4. In section 4.1 (Statistical analysis), the authors only handled the data within the descriptive statistics. In my opinion, ‘descriptive statistics’ is closer to ‘data description’ than ‘statistical analysis’.
#5. In the ‘Fig. 11’, what does Q mean?
#6. In section 4.4, What can ‘Time’ mean? Can this be interpreted as the time since entry into Weaving Area, rather than simply the recording time?
#7. This study was conducted using early morning data (7:05 a.m. to 8:35 a.m). So, are the results of this study an interpretation of a specific time period or specific event (office-going hour or rush hour)? Or can it explain general traffic characteristics?
#8. Is there any information about speed in this data? I think speed will have a big impact on things like crossing lanes.
Comments on the Quality of English Language○ Minor checkpoints.
- (L41-L76) The structure within the paragraph is too similar (opinion + ‘for example’ + reference).
- ‘Although these two LC behaviors have differentiated characteristics, simple analyses of the LC events were performed.’: I understand the author's intention, but the sentences are strange. What are simple analyses?
- (L283) downstream segments. , the weaving area is ~: what is ‘,’?
- (Fig. 8-9) Add the explanation for blue dot lines and purple box in the figure legend.
- ‘section 2.4’ in the manuscript -> section 3.4
Reviewer 2 Report
Comments and Suggestions for Authors
In this paper, the authors propose an approach using vehicle trajectory data to investigate the spatio-temporal characteristics of lane-changing at the freeway weaving area. The paper is well-organized, and the results are presented in a comprehensive manner. The manuscript is well organized, and the results enrich the existing theoretical research models and have certain theoretical meaning. Therefore, I think the paper can be accepted. In addition, the following are the few comments, which may be included while revision.
1. Lines 78-79, “the existing studies did not differentiate the mandatory and discretionary LC events that simultaneously occur at freeway weaving areas”. In fact, there are many studies differentiate the mandatory and discretionary LC events in highway weaving areas.
2. Lines 164-166, “In this study, LC events are classified into four types according to the LCVs’ origin-destination and LC directions, namely through vehicle inside LC event, through vehicle outside LC event, merging vehicle LC event, and diverging vehicle LC event”. How exactly these four types are divided is suggested to be represented graphically to facilitate the reader's understanding.
3. Lines 167 to 179 mainly summarize the research results related to void occupancy, and it is recommended that the authors place them uniformly in the introduction.
4. Line 212,“As seen in Fig. 6 (a)...”The author should first explain what the source of Figure 6 is, how it was obtained, and what its conditions and background are, rather than directly analyzing it.
5. Lines 449-450, “The curves in Fig. 11 fluctuate considerably, which means their slops could reflect the throughput variations very well”, why ? The author should provide a detailed explanation.
6. The author should emphasize the practical significance of these statistical trajectory data in the analysis, rather than just statistical data.
7. The list of references should be extended to include some recent papers as follow.
1) Performance-guaranteed fractional-order sliding mode control for underactuated autonomous underwater vehicle trajectory tracking with a disturbance observer.Ocean engineering, 2022.
2) Research on the Edge Resource Allocation and Load Balancing Algorithm Based on Vehicle Trajectory.Complexity, 2022, 2022.
3) A multivalue cellular automata model for multilane traffic flow under lagrange coordinate.Computational and Mathematical Organization Theory. 2022, 28. https://doi.org/10.1007/s10588-021-09345-w
4) Learning Vehicle Trajectory Uncertainty[J].ArXiv, 2022, abs/2206.04409.DOI:10.1016/j.engappai.2023.106101.
Reviewer 3 Report
Comments and Suggestions for Authors
The research work on evaluating the spatio-temporal characteristics of lane changes in highway weaving areas based on trajectory data has been carried out in this paper. The research content and results are very meaningful. Before publication, there is a small suggestion to add a discussion section.
Comments on the Quality of English Language 本文的语言非常流程Author Response
Please see the attachment.

Reviewer 4 Report
Comments and Suggestions for Authors
The present study aims to investigate the spatial and temporal distributions of lane-changing events and the corresponding disturbance on the freeway weaving. After reviewing the paper, the paper needs to address several key issues, which are as follows:
1. The introduction of the study could significantly benefit from an expanded literature review. A more detailed and focused discussion on lane-changing phenomena and methodologies from previous studies would provide a stronger, more comprehensive background, enhancing the paper’s contextual relevance.
2. There is a discrepancy in the data collection timeframe. The paper states the data was collected over 45 minutes, yet it also mentions a recording window from 7:05 am to 8:35 am. This inconsistency needs clarification to ensure the credibility of the data collection process.
3. The methodology used in this paper is simple. Also, the paper requires further justification for the chosen methods. It lacks a robust explanation of why these specific methods were selected and how they are best suited for this study.
4. The presented analysis is not comprehensive. A more in-depth examination of the causality between different types of LC events and traffic flow characteristics while considering the spatiotemporal effects would be more useful. Advanced statistical or machine learning techniques could be used to uncover more complex relationships.
5. While the study provides insights into LC events on the US Freeway, its scope for generalization to other freeways or traffic conditions remains unclear. The paper would benefit from a discussion on the applicability of its findings to different freeway environments or varying traffic scenarios.
6. While the study analyzes the spatial and temporal patterns of LC events, it could further explore the practical implications of these findings for traffic management and safety. How can this research inform better traffic control strategies in freeway weaving areas?
Comments on the Quality of English LanguageModerate editing of English language required.
Round 2
Reviewer 1 Report
Comments and Suggestions for Authors
The authors fully revised the manuscript reflecting comments of reviewers. However, I found a few things that needed modification. Please check these points. Authors will be able to revise their manuscript quickly.
○ Questions
#1. In the abstract, there is no reason to denote Origin-Destination with the abbreviation OD and LC vehicles’ with the abbrevitation LCVs’.
#2. I can understand the authors' intention, but I felt that the introduction was long and verbose (approximately 2.5 pages). If it's not an essential example, I think authors can shorten it a bit.
#3. Overall, the legend does not provide a description of the components within the figure. Regardless of the content of the main text, the figure must be understandable based on the figure legends alone. (e.g., the meaning of value in Fig. 8, ‘Q’ meaning in Fig. 11, purple boxes in Fig. 13)
#4. In coverletter, the authors explained to me why ‘general traffic characteristics’ can be explained by used data in this manuscript. I think that this reason explained by the authors should be included in this manuscript (comments 7 in author response file).
#5. (L667) Coifman and Li, 2017;) -> remove ‘;’.
Comments on the Quality of English Language-
Reviewer 4 Report
Comments and Suggestions for Authors
Thank you for addressing the comments provided on the earlier version of the paper. I have no further comments.
Comments on the Quality of English LanguageMinor editing of English language required.
Author Response
Thank you very much for your earlier comments.
I have checked the English language of the manuscript thoroughly this time.
